# Migrants and Healthcare during COVID-19, the Case of Kanchanaburi Province in Thailand

**DOI:** 10.3390/healthcare11202724

**Published:** 2023-10-13

**Authors:** Uma Langkulsen, Portia Mareke, Augustine Lambonmung

**Affiliations:** 1Faculty of Public Health, Thammasat University, Rangsit Campus, Bangkok 12120, Pathum Thani, Thailand; a.lambonmung@gmail.com; 2Biomedical Research and Training Institute, Harare P.O. Box A178, Zimbabwe; marekep06@gmail.com; 3Tamale Teaching Hospital, Ministry of Health, Tamale P.O. Box TL 16, Ghana

**Keywords:** social determinants of health, migration, health insurance policy

## Abstract

Since the outbreak of COVID-19, as reported by the WHO in December 2019 and subsequently declared a public health emergency of international concern, a distinct set of risk factors and vulnerabilities faced by migrants are affecting their exposure to the pandemic and its associated outcomes. This study aims to analyze the social determinants of health among migrants and their associated factors and compare the socio-demographic characteristics, patterns of COVID-19, and healthcare attendance and utilization among migrant workers and non-migrants. A descriptive study design was used to analyze COVID-19 morbidity among migrant workers. There were a total of 73,762 migrants living in the province by December 2021, with varied statuses and nationalities. Most of the migrants were from Myanmar, constituting about 80.1%. A total of 24,837 COVID-19 cases in Kanchanaburi province were recorded in 2020–2021. COVID-19 cases among migrant workers accounted for 22.3% during the period under review. Half, 2914 (52.7%) of the migrant female workers were victims of COVID-19 infections. Persons under the age of 18 accounted for about one-fifth of all the COVID-19 cases. Older, over 60 years old, Thais had about twice (10.1%) the COVID-19 cases compared with the older migrants (5.5%). There was a significant increase in healthcare attendance and utilization by non-migrants and migrants during the year under review. Migrants are at high risk of COVID-19 infection. Therefore, public health guidance for the prevention of COVID-19 should prioritize safeguarding the health of migrants by considering their individual characteristics and actions. Enhancing health insurance schemes for migrants, particularly vulnerable migrant groups, is critical for inclusive and expanded healthcare access. Physical, social, and economic environments that impact the health and well-being of migrants should be integral to pandemic prevention, preparedness, response, and recovery.

## 1. Introduction

The COVID-19 infection, formerly known as ‘2019 novel coronavirus’ or ‘2019-nCoV’, was originally discovered among cases of respiratory illness in Wuhan city, the epicenter of the outbreak in the Hubei province, China, and was reported to the World Health Organization (WHO) on 31 December 2019 [1]. The WHO classified the novel coronavirus (COVID-19) outbreak as a public health emergency of international concern (PHEIC) in January 2020. The WHO designated the respiratory illness brought on by a novel coronavirus known as severe acute respiratory syndrome coronavirus 2 (SARS-CoV-2) as COVID-19 in February 2020. By March 2020, there were numerous confirmed cases and fatalities, and it was declared a global pandemic by the WHO [1,2]. The pandemic continued to wreak havoc on individuals and nations until an emergency use authorization (EUA) and conditional permission for the first COVID-19 vaccines developed and made accessible to the public in 2020 [3]. Although efficient mRNA and viral vector vaccines have been developed quite quickly, global vaccine equity has not been achieved. High-income countries (HICs) receive the COVID-19 vaccine first, whereas low-income countries (LICs) must rely on voluntary donations through the COVID-19 vaccine global access (COVAX) [4]. Also, the creation and application of whole inactivated virus (WIV) and protein-based vaccines with seemingly different efficacy, particularly for use in impoverished nations, is a concern [5,6]. Nevertheless, the COVID-19 vaccinations are highly recognized for their contribution to decreasing COVID-19 severity and fatality rates as well as significantly helping to contain its spread. It has been reported that the COVID-19 vaccination averted an extra 14.4 to 19.8 million deaths in 185 countries and territories. According to official data from national public health organizations, over 13.2 billion doses of the COVID-19 vaccine have been administered globally as of February 2023 [1]. Many nations adopted staggered vaccination plans that gave priority to people who were at higher risk of developing complications and dying, such as the elderly, and transmission, like healthcare personnel, but still had to overcome an issue of vaccine apathy in some instances [4].

The COVID-19 pandemic had a significant impact on daily lives, businesses, and global trade. The global economy and financial markets, as well as the daily lives of people, have significant impacts, including social, economic, and educational [6]. Vulnerable and disadvantaged groups, including some ethnic minorities, migrants and those with low socioeconomic status (SES), have been disproportionately affected by COVID-19 spread across borders and within nations [2,7,8,9,10]. The health gaps between these groups, which are fueled by intricate socioeconomic factors and enduring structural imbalances, have been revealed during the COVID-19 pandemic.

Global migration was probably highest before the COVID-19 pandemic, with 180.5 million people moving around the globe, affecting the employment prospects of refugees and people seeking asylum [9]. The International Organization for Migration (IOM) broadly defines a migrant as any person who is moving or has moved across an international border or within a state away from his/her habitual place of residence, regardless of the person’s legal status, whether the movement is voluntary or involuntary, what the causes for the movement are, or what the length of the stay is [11]. Large-scale migration of people within countries and across borders is influenced by the complex interaction of a variety of factors, making their unique healthcare needs a special global health challenge. Lack of data, as well as inadequate communication between healthcare providers and patients, can result in a misunderstanding of healthcare demands [12,13]. The situation is often made worse by the difficulties that migrants encounter in exercising their human rights, gaining access to healthcare and other basic services, and being forced into low-paying, frequently dangerous jobs [14,15]. To this end, 193 UN Member States have ratified the New York Declaration for refugees and migrants to address the pandemic risks and the unique health challenges, as well as to provide access to basic needs for health, education, and psychosocial development [15]. Although a picture is starting to take shape of how much the COVID-19 pandemic has affected migrants, the immense increase in migration both within countries and across borders and the new public health opportunities and challenges are becoming apparent. This could be attributed to the sizable number of migrant frontline workers who may be more exposed to COVID-19 and/or operate in industries where COVID-19 infections have a disproportionately negative effect on migrants [2,16].

This study evaluated COVID-19 infection among migrants and non-migrants and analyzed it in relation to social determinants of health among migrants. The available health services and the public health guidance for the prevention of COVID-19 and safeguarding the health of migrants in Kanchanaburi province, Thailand, were also analyzed. The descriptive data would be useful in enhancing the health of migrants, particularly groups like females, children, stateless, and undocumented people in vulnerable situations, as well as in helping to inform public health practice and policy decision-making by practitioners.

### 1.1. Migrants in Thailand

Thailand is regarded as an attractive destination for migrants. The Migration Policy Institute (MPI) ranks Thailand 16th in its top 25 migrant destinations globally, with over 3.5 million international migrants, or about 5.2% of the country’s total population [17]. The Kingdom of Thailand contributes significantly to regional and international migration as an origin, destination, and transit country. The nature of economic growth in a more globally interconnected world has boosted the contribution of international migration (refugees, internally displaced people, professionals, and labor migrants) in Thailand’s economy. The Kingdom has been effective in recruiting a lot of employees from overseas to work. Thailand’s economy is comparatively strong and stable, which has drawn millions of migrants from its neighbors seeking a higher quality of living. Many industries, including fishing, agriculture, hospitality, household services, and manufacturing, significantly rely on migrant labor. Three nations account for the great majority of the migrants in Thailand: Cambodia, the Lao People’s Democratic Republic, and Myanmar [18]. Additionally, the nation has a history of hosting tens of thousands of migrants from its neighbors who had to flee their homes because of war, civil unrest, or other national instability. Currently, nine camps on the Thai-Myanmar border are home to an estimated 105,000 refugees [19].

Thailand has made significant progress in defending the rights of migrants while appreciating their contributions in the 30 years since it became a member state of the International Organization for Migration. The Royal Thai Government has recently developed its own migration management strategy using a pragmatic and creative approach. In this regard, bilateral memoranda of understanding (MoUs) in the areas of counter-trafficking and labor migration have been inked with surrounding nations [20]. However, with about 1 million of the estimated 4 million migrants reported to be residing and working in Thailand believed to have irregular status, the government’s main concern with regard to migration is irregular migration, particularly people-smuggling and human trafficking, and its effects on the labor market and public health [19].

Unsurprisingly, Thailand was the first nation outside of China to report a COVID-19 case detected in a Wuhan-based traveler to Bangkok in January 2020 [21]. Despite this, the country has been commended for its response to the waves of COVID-19 infections, especially in protecting the health and well-being of migrants in the country. During the first wave of COVID-19, which started in March 2020, clusters of cases connected to activities at the Bangkok boxing stadium and nightclubs were reported. The infection was also introduced to the southern region of Thailand by Muslim pilgrims who were traveling back from Malaysia and Indonesia, which ultimately spread to 68 provinces in the country. Epidemiological data showed that there was no local transmission until after 25 May 2020, since all cases were discovered in non-Thai and Thai travelers at state quarantine systems [22,23]. Several Thai laborers who had worked at an amusement park in a northern state of Myanmar started the second wave but managed to enter Thailand without being apprehended by state quarantine. They transported the virus and dispersed it to Thailand’s northern districts. Also, a sizable number of migrants with the virus arrived directly and illegally from Myanmar [23].

### 1.2. Social Determinants of Health and Migrants

Despite more widespread access to medical treatment, there are still significant socioeconomic class disparities in health across several nations among migrant groups [24]. Health inequalities, or the unjust and preventable variations in health status found within and across countries, are significantly influenced by SDH, which is mostly unfavorable to migrants [25]. Health and sickness follow a social gradient across nations of all income levels: the poorer one’s socioeconomic standing, the worse one’s health, as has been badly exposed by the COVID-19 pandemic, particularly in low socio-economic migrant communities [26]. The social determinants of health are examples that can have both good and negative effects on health equity. Aiming towards the highest level of health for everyone while paying particular attention to the needs of those most at risk of ill health due to socioeconomic circumstances is what it means to pursue health equity. Calls for action for both working beyond the healthcare system to address broader social well-being and the development of migrant communities to reduce their vulnerabilities to pandemic outbreaks [21]. All sectors, including civil society, must make efforts to properly address SDH in order to improve health and reduce long-standing health disparities, especially among migrants [24,27].

In many nations, immigrants may not have the same access to healthcare as citizens, especially if they are of irregular status or have temporary visas; therefore, they may not be covered for COVID-19 treatment, exacerbating their health inequity [28]. In an initial assessment of the current pandemic, the IOM warns that societies that do not adequately guarantee health care, aid, and access to fundamental rights to such broad population groupings will be less able to effectively limit the outbreak, will probably see a higher overall number of people affected and will likely experience an emergency situation that lasts longer [29]. Irregular migrants may be reluctant to come forward if they are afraid of being reported to immigration officials and deported if they seek help. It can make them less reluctant to participate in screening, testing, finding contacts, or receiving treatment. Additionally, language hurdles, a lack of familiarity with the host culture, or the prioritizing of residents may prevent individuals from receiving the appropriate services even though they are entitled to them. In addition to individual-level adjustment processes to a new environment, migration also involves adaptation difficulties involving the complex and frequently drawn-out process of negotiation in social structure and political and economic forces that can significantly impact their health and well-being [30]. The scope of social and health disparities for migrants has particular serious repercussions for their health and well-being [17]. Tens of thousands of migrants are often accommodated as high risk for COVID-19 at camps, detention facilities, and labor dormitories or compounds [26]. Some migrants have suffered a greater risk of severe COVID-19 diseases, which partly is due to the function of the social and economic circumstances [26,31,32]. During the COVID-19 epidemic, refugees and migrants who experienced heightened discrimination and high levels of stress related to their basic material and medical requirements reported seeing a marked decline in their mental health. Furthermore, migrants and refugees who have less secure housing and residency status are more likely to experience mental health issues [33].The challenges mentioned by migrants are with communication barriers, experiencing loneliness due to separation from the social support systems, not receiving health interventions compared to the general populations, and lack of access to information [21,26].

Although migration is socially driven, to the extent that immigration and immigrant integration into society have an impact on many social and economic aspects that affect health, including economic stability, access to healthcare, education, the physical environment’s effect, and social and communal context, it is also regarded as a social determinant of health [12,13]. In this regard, the 61st World Health Assembly passed a resolution encouraging nations to create health practices and policies that are considerate of migrant populations [34]. Also, the areas and nations with sizable populations of migrants are admonished to make sure their needs are better considered when developing public health strategies to improve the migrant health [35,36]. These emphasize the importance of immigrants in the social, cultural, and economic fabric of our globalized world, suggesting that only inclusive approaches can help protect and promote everyone’s rights, health, and well-being [12]. Such an approach would enable communities and societies to respond to the COVID-19 crisis more effectively and substantially reduce the impact of pandemics on migrants in the future.

### 1.3. Public Health Measures for COVID-19 in Thailand

The Ministry of Public Health (MOPH) is responsible for the implementation of Universal Health Coverage (UHC) since the latest reorganization of healthcare delivery in the country in 2002. Under the supervision of the ministry, operating in a five-tier system, from central, regional, provincial, district, and sub-district levels, healthcare is provided with insurance to ensure expanded access for all [37]. In partnership with the private sector, health insurance coverage is provided for migrants as well as the indigenous population [38].

Though Thailand was the first nation outside of China to report a COVID-19 case, the country experienced a 102-day period between May and September 2020 without any local transmission of COVID-19 being recorded after an initial rise in cases. The public health response in Thailand was prompt and thorough. By isolating and treating confirmed individuals as well as locating and quarantining their connections, rapid response teams effectively controlled confirmed cases. Instead of being treated at home, all patients received isolation in institutions. By the end of July, 78% of Thailand’s 77 provinces had the ability to diagnose COVID-19 thanks to the establishment of a laboratory network for RT-PCR diagnosis. Volunteers and migrant health workers were used to mobilize support for social and public health initiatives. Selective and targeted interventions were implemented throughout the provinces with various social measures, including face masks, physical distance, and hand hygiene [39,40]. Due to Thailand’s four decades of investment in its healthcare system, the nation is well-positioned to address the present public health issue [41].

The guidelines for the surveillance implementation for control and prevention of COVID-19 among migrant workers included surveillance measures for target group 1: patients under investigation (PUI); target group 2: irregular migrant workers or migrant workers who violate laws in Thailand; and target group 3: migrant workers who live in slums or work in facilities in such areas determined to have high exposure to COVID-19 in the province. The Department of Disease Control (DDC) and the DDC-COVID-19 reporting system issued codes (SAT Code) representing cases under investigation for COVID-19. SAT code M was issued for migrant workers. The number of migrant workers in the surveillance system was approximately 46,200 and 26,919 for target groups 1 and 2 migrant workers per year, respectively. For target group 3, the number of migrant workers in the surveillance system was approximately 25,940 annually [42]. The submission of specimens for COVID-19 testing for target groups depended on the COVID-19 situation among the migrant workers. Specimen collection methods to test for COVID-19 were collected with a nasopharyngeal swab and labeled in a viral transport media tube (VTM) or universal transport media tube (UTM) with the ID code of the patient and date of collection in accordance with the Ministry of Public Health’s COVID-19 pandemic response guideline’s details on specimen collection, specimen preservation, and transportation. International statistical classification of diseases and related health problems, 10th revision (ICD-10) was used to record confirmed COVID-19 cases and comorbidities. While the ICD-10 code U07.1 was used to define the COVID-19 virus identified, ICD-10 codes J02.8 and J12.8 were used to define acute pharyngitis due to other specified organisms and other viral pneumonia as comorbidities when a person is diagnosed as having COVID-19 acute pharyngitis and COVID-19 pneumonia, respectively [43]. Budget disbursement for the laboratory testing for COVID-19 is divided into two categories: migrant workers who already have the Health Insurance Card Scheme (HICS) and migrant workers without access to insurance schemes. While the cost of COVID-19 testing was directly disbursed through the Department of Disease Control, migrant workers who are not registered under any insurance scheme were processed by government agencies under the Department of Disease Control, the Department of Medical Sciences, the Office of the Permanent Secretary, and the Ministry of Public Health, in accordance with the migrants healthcare under the Universal Health Coverage [44].

## 2. Materials and Methods

### 2.1. Study Area

Kanchanaburi province, as shown in Figure 1, is the third largest of the 77 provinces, covering an area of 19,473 km^2^ and located in the lower central region of Thailand. It is one of the ten specific Special Economic Zones (SEZs) consisting of Tak, Sa Kaeo, Trat, Mukdahan, Songkhla, Chiang Rai, Nong Khai, Nakhon Phanom, Narathiwat, and Kanchanaburi provinces. In the west, it borders Myanmar’s Tanintharyi region, Kayin, and Mon States [45].

### 2.2. Study Population

The number of migrants considered for the study was 24,481 (migrants living in the province by the end of 2021). This included migrants under Section 59 who are temporarily permitted to stay in Thailand according to the Cabinet Resolution of 2019 and have completed their nationality verification. Migrants who migrate to work in Thailand under the Memorandum of Understanding between the government of Thailand and partners [46]. It also includes migrant workers under Section 63/2 who enter Thailand without permission according to the immigration law but are granted temporary stay permits while waiting to be processed for deportation. Migrant workers under Section 64 who work in the border areas as daily or seasonal workers under the Agreement on Border Crossing between Thailand and the neighboring countries (Lao PDR, Myanmar, and Cambodia) [47] were also considered. Undocumented migrants in the province were not included, as the study largely relied on secondary sources of data that did not have specific information on this group.

### 2.3. Instrument

The descriptive study design was used to gather data on migrants and social determinants of health in relation to the COVID-19 pandemic to assess COVID-19 morbidity and healthcare access by migrants in the Kanchanburi province of Thailand. This is an important tool for determining the prevalence of the pandemic and its associated factors among this population group during the study period in the province [48].

### 2.4. Data Collection and Analysis

Permission was given by the provincial health authorities for the extraction of health data from the Health Data Centre (HDC) database for Kanchanaburi province and the Department of Disease Control/COVID-19 control. It contains data on health services in the out-patient department (OPD), in-patient department (IPD), and medical expenses during the period of 2017–2021 in the province. The epidemiological situation of COVID-19 during 2020–2021 was collected from the Department of Disease Control. As per the guidelines of The Human Research Ethics Committee of Thammasat University (Science), Thailand (HREC-TUSc) (COA No. 115/2562), and the Institutional Review Board (IRB) for the study, data protection and patient confidentiality were ensured. The descriptive summary statistics of the healthcare attendance and utilization were presented in the form of tables and a graph. Additionally, to compare these aspects between non-migrants and migrants, a chi-square test was performed to find the statistical level of significance using IBM SPSS Statistics, version 28 (SPSS28).

## 3. Results

### 3.1. Demographic Data

The number of total migrants in Kanchanaburi province was 73,762 by 2021 December, as shown in Table 1. The majority of the migrants were from Myanmar 59,107 (80.1%). The dominance of Myanmar migrants in the province was expected, as it has been the case in the entire country for decades. Aside from proximity, conflicts and political instability have been the main drivers [19], followed by others (undetermined nationality) at 12,005 (16.3%). Vietnam was the country with the fewest migrants in the province, with 7 (0.01%). The year 2017 recorded the lowest number of migrant residents (41,843). In 2021 (73,762), the number of migrants living in the Kanchanaburi province of Thailand was the highest during the five-year period.

### 3.2. COVID-19 Morbidity

During the period under review, a total of 24,837 COVID-19 cases were recorded among migrants and non-migrants (Thais) in Kanchanaburi province. COVID-19 cases among migrant workers accounted for 22.3% during the period under review. The majority of the COVID-19 cases among the migrant workers were from Myanmar (99.0%), followed by Cambodia (0.7%) and Lao PDR (0.3%). There was no confirmed COVID-19 infection among migrants from Vietnam. Over half of 2914 (52.7%) migrant female workers were victims of COVID-19 infections, similar to non-migrant females, 9912 (51.3%). Persons under the age of 18 accounted for about one-fifth of all the COVID cases. Older Thai people, over 60 years old, were about twice (10.1%) COVID-19 cases compared with the migrants (5.5%). Migrants were significantly associated with COVID-19 (22.3%). There was no statistically significant sex difference between the non-migrant and migrant workers (*p*-value > 0.05). However, there were statistically significant age-related differences (*p*-value < 0.05). The different socio-demographic characteristics of confirmed COVID-19 cases among migrants and non-migrants are summarized in Table 2.

### 3.3. Utilization of Health Services

The study also determined access to healthcare services and per capita costs for migrants during the COVID-19 period of 2020–2021. Data covering migrant workers and members of their dependents showed a significant difference in the year before the COVID-19 outbreak and years after in the utilization of health services in the province. As shown in Table 3, the number of annual migrant healthcare visits in the Kanchanaburi province increased drastically after the province recorded COVID-19 cases in the year 2021 (85,025) compared to the non-COVID-19 years. There was a significant difference in healthcare attendance and utilization in the 2021 and 2020 in the province. During this period, there was an increase of 30.6% by migrants compared to non-migrants 7.9% in healthcare service utilization.

The number of migrant healthcare visits and utilization, including health expenditure per capita in the year 2021, increased markedly. The rise in health services utilization resulted in a corresponding significant rise in health expenditure in 2020 (131) and 2021 (215) (USD) per capita, as presented in Figure 2 below. Per the current healthcare financing scheme for migrants in Thailand [49], about 8% of the health expenditure incurred was financed via the health insurance schemes. The remaining 92% (free services) was borne by the government through the Ministry of Public Health [38].

## 4. Discussion

### 4.1. Migrants and Vulnerabilities to COVID-19

The substantial number of COVID-19 infections (22.3%) among migrants found could be attributed to the province’s host of large migrant communities. Substandard socio-economic conditions of migrants have been determined to be associated with a high incidence of communicable diseases such as diphtheria, pulmonary tuberculosis, malaria, syphilis, cholera, leprosy, and lymphatic filariasis in many countries [50,51,52,53,54]. On the current pandemic, various studies have reported similar findings of high recorded COVID-19 cases in migrant communities [10,55,56]. The vulnerabilities of migrant populations are probably largely due to inequities in some factors that influence health and the fact that migration itself is determined to be a health determinant [30]. Due to the uniqueness of migrants in terms of their individual characteristics and actions and their peculiar physical, social, and economic environments, they are often vulnerable to diseases [11,12,30]. Similar to this study, Liu et al. found high COVID-19 cases in migrant workers from some particular places in China. To avoid forced eviction and subsequent troubles due to discrimination during the pandemic, they tended to conceal COVID-like symptoms, fueling an increase in the risk of infection within their neighborhoods and workplaces [57]. Likewise, Fabreau et al. have reported that migrant workers in meat processing plants have to perform physically demanding work in noisy environments that requires shouting to communicate, thereby increasing their risk of infection [58]. Some studies found migrant workers in Thailand to have been disproportionally affected by COVID-19 as a result of poor social and economic conditions [7,22,59]. The subpar condition of camps, inadequate facilities, and meager meals are the major stated reasons for the high cases of COVID-19 among migrants [2,8,60,61,62]. More so, and probably the most impacted continent of Europe, a systematic review found migrants to be the most affected by the COVID-19 infection [63]. Another review concluded that Latinos, African Americans, and other minority groups in the United States were more likely to be affected by the pandemic. The conclusion was on the basis that the pandemic was more of a social problem, particularly in these groups that have many and large migrant communities in the country and live in poor socio-economic conditions [10]. This supports the findings of this study, suggesting that poor socio-economic factors of migrants could have contributed significantly to their vulnerability to the COVID-19 infection in the province. Gender-sensitive policies for combating COVID-19 were proposed to cater to migrant women’s multiple vulnerabilities [8]. The gender disparity of the migrants in the province of 50.6% (females) could have been a reason for the higher number of COVID-19 infections among female migrants, as females have been reported to be more exposed to COVID-19 infection [2,31,64]. A study in India attributed the poor socio-economic conditions of migrants to be responsible for women’s susceptibility to COVID-19 infection [62]. The negative correlation of the spread of COVID-19 with variables relating to substandard housing facilities that are often associated with migrant communities because of the poor socioeconomic conditions is likely a key factor, as has been reported variously [10,26,56]. Another link is that economically, vulnerable and excluded groups such as migrants find it difficult to adhere to the social segregation policies put in place by states and municipalities because they need to continue working to ensure their survival, as reported by some COVID-19-related studies [10,21,64]. Migrants reported a substantial difference in the incidence and mortality rates of COVID-19, which illustrates the disparities in the experience of the pandemic and indicates that migration as a social determinant of health could have significantly contributed to the dissemination of COVID-19 in Thailand and in particular Kanchanaburi province, which has a sizeable migrant population [60].

### 4.2. Healthcare Access and Utilization

Regarding the utilization of healthcare services by migrants, the study observed a high record of attendance, particularly during the 2021 COVID-19 period. Although studies have reported decreased healthcare access as a result of COVID related restrictions and barriers relating to communications [65,66]. Other vital services, like mental healthcare, were also impacted [67]. Due to the peculiar challenges for migration and health, including access to healthcare, health service delivery, and healthcare financing, together with the often appalling living conditions of some migrants, COVID-19 could have been an exacerbation of a health crisis. COVID-19 could have been a socioeconomic problem and a protection crisis for some migrants in the country [36,59,64], but for Thailand’s distinguished immigrant health policy framework [41]. Thailand’s migrant health insurance scheme [68], which provides funding assistance for migrant health challenges, is regarded as exemplary as it offers migrants increased access to healthcare [40,59]. The expanded testing and treatment facilities for migrants and increased use of healthcare services by migrants in Kanchanaburi province for COVID-19 could have been facilitated by the Universal Health Coverage Policy [69]. Increased utilization of healthcare services due to COVID-19 among migrants has been determined to be high in territories and nations with health policies and programs favorable to migrants [65,70]. The IOM, in its initial assessment of the pandemic, indicated that the patterns of vulnerability are unique to migrants and that the patterns of the first and second waves of illnesses could make migrants more vulnerable victims of the pandemic, and it recommended expanded access to primary healthcare for migrants [71]. Access to health care through innovative ways in order to organize translating and volunteer services across several practices in primary health care was recommended by migrants and healthcare providers to help contain the disease among migrants through a qualitative inquiry [70], a feature that is promoted by Thai health authorities, including free and open access to facilities at the border region for migrants [38]. Though more needs to be done to expand access to undocumented migrants and stateless people [68]. The general health support programs for immigrants in Thailand could be a reason for the increased coverage of healthcare services for migrants before and during the pandemic. This possibly explains the low COVID-19-related morbidity and mortality among migrants in Thailand as compared to other nations that host high numbers of migrants. It emphasizes how important it is to consider migrants’ health in the design and implementation of public health policies and interventions.

Since it is often challenging to have access to accurate and current data on migrants, this could be a limitation of the study. We, however, relied on the efficiency of the Health Data Centre database in the province. In addition, since the study used data on migrants from one province, it limits its generalization.

## 5. Conclusions

In conclusion, the findings imply that socioeconomic variables among migrants could have played a substantial role in their exposure to COVID-19-related morbidities. To this end, targeted health policies that aim at expanding access to healthcare delivery for migrants are critical to fulfilling the rights to health and well-being of migrants, especially in times of crisis. Integration and inclusion of migrants, with a focus on enhancing their socioeconomic conditions, particularly migrant groups like females, children, undocumented, and the stateless, are critical, as involuntary or voluntary migration would probably continue to exist. Therefore, solutions relating to social determinants of health and migrant health policies, especially health insurance, that make migrants part of society and structural rather than external in health systems and other areas, are crucial for increasing access to essential services like health care during this pandemic and in the future.

## Figures and Tables

**Figure 1 healthcare-11-02724-f001:**
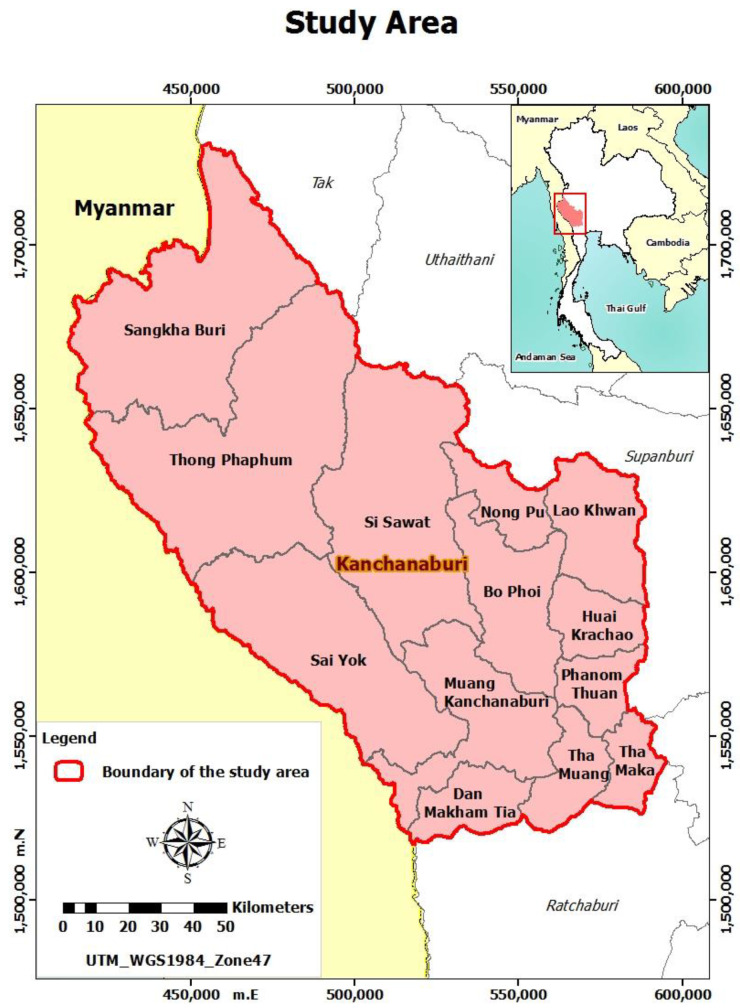
Study area.

**Figure 2 healthcare-11-02724-f002:**
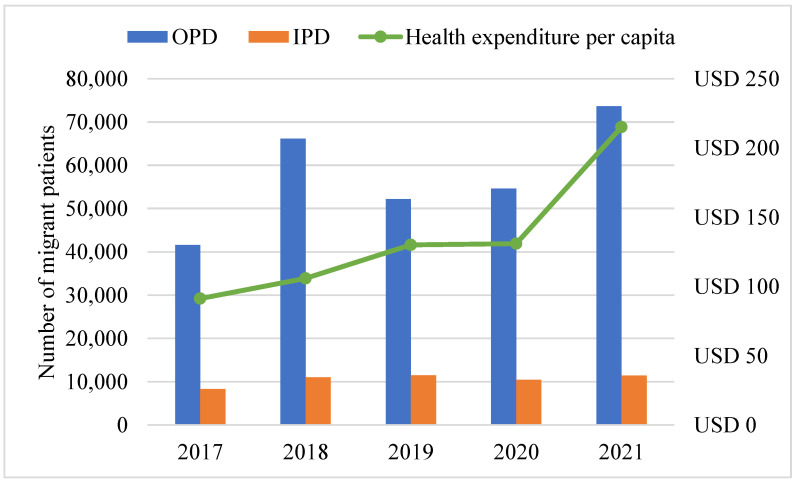
Number of migrant outpatient visits and inpatient admissions and health expenditure per capita in Kanchanaburi province from 2017 to 2021. Source: Ministry of Public Health.

**Table 1 healthcare-11-02724-t001:** Characteristics of migrants in Kanchanaburi province, 2017–2021.

Migrants in Kanchanaburi Province					
Characteristics	2017 (*n* = 41,843)	2018 (*n* = 66,222)	2019 (*n* = 52,369)	2020 (*n* = 54,860)	2021 (*n* = 73,762)
Country of origin, *n* (%)					
Myanmar	31,383 (75.0)	50,157 (75.74)	37,231 (71.09)	42,102 (76.8)	59,107 (80.13)
Lao PDR	1006 (2.4)	1528 (2.3)	1355 (2.59)	1328 (2.4)	1527 (2.07)
Cambodia	492 (1.18)	1549 (2.34)	780 (1.49)	1006 (1.8)	1116 (1.51)
Vietnam	9 (0.02)	11 (0.02)	18 (0.03)	12 (0.02)	7 (0.01)
Others	8953 (21.4)	12,977 (19.6)	12,985 (24.8)	10,412 (18.98)	12,005 (16.28)
District in Kanchanaburi province, *n* (%)					
Mueang Kanchanaburi	2226 (5.32)	14,082 (21.26)	9711 (18.54)	11,452 (20.88)	17,303 (23.46)
Sai Yok	6199 (14.82)	8222 (12.42)	6613 (12.63)	6720 (12.26)	6964 (9.44)
Bo Phloi	935 (2.23)	1976 (2.98)	982 (1.88)	1401 (2.55)	2733 (3.71)
Si Sawat	961 (2.3)	1299 (1.96)	1071 (2.05)	1204 (2.19)	1566 (2.12)
Tha Maka	5127 (12.25)	6528 (9.86)	6830 (13.04)	5460 (9.95)	8523 (11.55)
Tha Muang	3433 (8.2)	4383 (6.62)	2327 (4.44)	3439 (6.27)	3183 (4.31)
Thong Pha Phum	11,679 (27.91)	14,145 (21.36)	11,160 (21.31)	13,316 (24.27)	15,170 (20.57)
Sangkhla Buri	9735 (23.27)	11,199 (16.91)	11,192 (21.37)	8925 (16.27)	11,768 (15.95)
Phanom Thuan	649 (1.55)	2125 (3.21)	1170 (2.23)	1492 (2.72)	1385 (1.88)
Lao Khwan	74 (0.18)	496 (0.75)	285 (0.54)	329 (0.6)	498 (0.68)
Dan Makham Tia	540 (1.29)	1151 (1.74)	496 (0.95)	473 (0.86)	4152 (5.63)
Nong Prue	183 (0.44)	447 (0.68)	293 (0.56)	329 (0.6)	188 (0.25)
Huai Krachao	102 (0.24)	169 (0.25)	239 (0.46)	320 (0.58)	329 (0.45)

Source: Ministry of Public Health.

**Table 2 healthcare-11-02724-t002:** Comparison of COVID-19 cases among migrant and non-migrant workers according to different socio-demographic characteristics.

Characteristics ^1^	Non-Migrant Workers (*n* = 19,309)	Migrant Workers (*n* = 5528)	Total (*n* = 24,837)
Sex, *n* (%)			
Female	9912 (51.3)	2914 (52.7)	12,826 (51.6)
Male	9179 (47.6)	2584 (46.8)	11,763 (47.4)
Unspecified	218 (1.1)	30 (0.5)	248 (1.0)
Age (years) *, *n* (%)			
<18	3952 (20.5)	1023 (18.5)	4975 (20.0)
18–24	2262 (11.7)	699 (12.7)	2961 (11.9)
25–34	3578 (18.5)	1188 (21.5)	4766 (19.2)
35–44	3020 (15.6)	891 (16.1)	3911 (15.8)
45–54	2637 (13.7)	607 (11.0)	3244 (13.1)
55–60	1153 (6.0)	200 (3.6)	1353 (5.4)
>60	1955 (10.1)	305 (5.5)	2260 (9.1)
Unspecified	752 (3.9)	615 (11.1)	1367 (5.5)

Source: Ministry of Public Health. ^1^ Chi-square test for categorical variables. * *p*-value below 0.05.

**Table 3 healthcare-11-02724-t003:** Comparison of healthcare attendance and utilization by non-migrants and migrants by year.

Healthcare Attendance and Utilization by Non-Migrants and Migrants								
Year	Non-Migrant Workers			Migrant Workers			X^2^	*p*-Value *
	Outpatient Care, *n* (%)	Inpatient Care, *n* (%)	Total	Outpatient Care, *n* (%)	Inpatient Care, *n* (%)	Total		
2021	1,090,155 (91.9)	95,492 (8.1)	1,185,647	73,601 (86.6)	11,424 (13.4)	85,025	2982.1	<0.0001
2020	999,162 (91.0)	99,199 (9.0)	1,098,361	54,656 (84.0)	10,429 (16.0)	65,085	3519.7	<0.0001
2019	1,052,305 (90.9)	105,765 (9.1)	1,158,070	52,222 (82.0)	11,478 (18.0)	63,700	5495.5	<0.0001
2018	1,003,469 (90.7)	102,624 (9.3)	1,106,093	66,120 (85.7)	11,037 (14.3)	77,157	2098.7	<0.0001
2017	905,752 (92.0)	78,736 (8.0)	984,488	41,581 (83.4)	8299 (16.6)	49,880	4599.1	<0.0001

Source: Ministry of Public Health. * *p*-value shows the significance between the two proportions of outpatient care between non-migrant and migrant workers.

## Data Availability

Not applicable.

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
