# Peer review of "Migrants and Healthcare during COVID-19, the Case of Kanchanaburi Province in Thailand"

_healthcare, 2023, doi:10.3390/healthcare11202724_

Round 1

Reviewer 1 Report (New Reviewer)

In general, this is an interesting article due to the size of the sample and the context it examines. However, it is felt that it could be improved by adding some extensions, mainly aimed at further detailing the method, as well as deepening the results by adding complementary analyses. Our proposals are presented below:

Introduction:

1)      You mentioned "vulnerable and disadvantaged groups". Please be careful with the terms you use to avoid blaming the victim. Perhaps "populations in a situation of vulnerability"? See also https://nccdh.ca/images/uploads/Population_EN_web2.pdf

2)      I would suggest addressing the social determinants of migrant health directly, integrating 1.1. and 1.2. by focusing on how COVID-19 has exacerbated these health inequities. This is now presented in two rather superficial sections, containing information that regular readers of this magazine are already familiar with. In this regard, please provide more citations related to how the pandemic has affected migrant health and, in particular, its determinants, from multilevel and contextualized approaches. Some references that may help in this regard are:

a.       MIPEX INTEGRATION POLICY INDEX (health strand):  www.mipex.eu

b.       Apart Together project: https://www.who.int/publications/i/item/9789240017924 , 10.1186/s12889-022-13370-y , 10.3390/ijerph18126354

3)      I would suggest adding a subsection to contextualize the study, explaining the reality of migration in Thailand with data, as well as the Thai healthcare system and how COVID-19 has affected both migrants and non-migrants in general. This information is now scattered across several sections (e.g., 2.1, 2.2), and the reader may get lost reading it without a clear framework of the picture.

Materials and Methods:

This section should be more detailed, with at least four sections:

2.1. Participants: with a clear description of the study participants and how they were selected (much of this information appears later in Results: 3.1) 3.2.

2.2. Instruments: with a detailed description of the instruments used for data collection.

2.3. Data collection and analysis. detailing how these data were collected and the ethical issues that were considered in doing so. Furthermore, detailing the types of analysis and how they were carried out.

Results

We recommend including further analysis to deepen the findings, as there is sure to be additional data that will address important gaps. For instance, how did gender influence service use? And what about socioeconomic or migration status? This would certainly enrich the discussion.

Discussion & Conclusions

The discussion is correct, although it lacks a more applied part, such as lessons for policy makers from the data obtained. Likewise, the limitations could be discussed in more depth (epidemiology is far from knowing the problems of migrants from their own voice), as well as the lines for the future that this study opens up in its context and beyond.

Author Response

Dear Reviewer 1,

The comments and observations of the reviewer are well appreciated. The specific responses are given below.

Comment

Introduction:

You mentioned "vulnerable and disadvantaged groups". Please be careful with the terms you use to avoid blaming the victim. Perhaps "populations in a situation of vulnerability"? See also

Response

Thanks for this observation. There are no deliberate intentions to blame the victims, and this has been corrected as suggested in page 3 line 41.

Comment

I would suggest addressing the social determinants of migrant health directly, integrating 1.1. and 1.2. by focusing on how COVID-19 has exacerbated these health inequities. This is now presented in two rather superficial sections, containing information that regular readers of this magazine are already familiar with. In this regard, please provide more citations related to how the pandemic has affected migrant health and, in particular, its determinants, from multilevel and contextualized approaches. Some references that may help in this regard are:

  1. MIPEX INTEGRATION POLICY INDEX (health strand): www.mipex.eu
  2. Apart Together project: https://www.whoint/publications/i/item/9789240017924, 10.1186/s12889-022-13370-y, 10.3390/ijerph18126354

Response

This is highly regarded as it has helped to reorganise the manuscript to be more focused on specific issues in each section. Section 1.2 (new) Social Determinants of Health (SDH) and migrants.

Address the SDH of migrants in Thailand in the context of COVID-19. The useful resource provided is cited in line 32 of this section.

Comment

I would suggest adding a subsection to contextualize the study, explaining the reality of migration in Thailand with data, as well as the Thai healthcare system and how COVID-19 has affected both migrants and non-migrants in general. This information is now scattered across several sections (e.g., 2.1, 2.2), and the reader may get lost reading it without a clear framework of the picture.

Response

This is equally an important input helping in the reorganization of the manuscript.

Section 1.1 Migrants in Thailand, pages 4 and 5 focuses on migration in Thailand and COVID-19

Comment

Materials and Methods:

This section should be more detailed, with at least four sections:

2.1. Participants: with a clear description of the study participants and how they were selected (much of this information appears later in Results: 3.1) 3.2.

Response

The study population in section 2.2 (Study Population) provides information on the participants in pages 6 and 7.

Comment

2.2. Instruments: with a detailed description of the instruments used for data collection.

Response

This is a very helpful comment. Section 2.3 Instrument provides information for the tool used in page 7, lines 12-18.

Comment

2.3. Data collection and analysis. detailing how these data were collected and the ethical issues that were considered in doing so. Furthermore, detailing the types of analysis and how they were carried out.

Response

Thanks for comment. This information is now presented in Section 2.4 Data Collection and Analysis in page 7, lines19-32.

Comment

Results

We recommend including further analysis to deepen the findings, as there is sure to be additional data that will address important gaps. For instance, how did gender influence service use? And what about socioeconomic or migration status? This would certainly enrich the discussion.

Response

The comment is well appreciated. Page 8 lines 24-27 and Table 2 provide further analysis of the results.

Comment

Discussion & Conclusions

The discussion is correct, although it lacks a more applied part, such as lessons for policy makers from the data obtained. Likewise, the limitations could be discussed in more depth (epidemiology is far from knowing the problems of migrants from their own voice), as well as the lines for the future that this study opens up in its context and beyond.

Response

Thanks for this critical observation. Since the study design elicited a more general and broad data, the discussions and the conclusions were borne out of the generalised context in which the issues were presented and discussed. However, some information relating to the above can be found in lessons for policy makers-page 12 lines 26-32, the future- page 12 lines 26-32, and limitations- page 12 lines 19-22.

Thank you very much.

Reviewer 2 Report (Previous Reviewer 3)

Thank you very much for revising the work.

Author Response

Dear Reviewer 2

Thank you very much.

Reviewer 3 Report (New Reviewer)

Use authors names, not citation numbers when citing them in the article. For example, line 325 states "49 reported...." Should say "Fabreau et al. (49) reported...."

Introduction should provide theme and purpose of the article, not jump right into the literature review.

More information needed to define a descriptive study design.

Author Response

Dear Reviewer 3,

The comments and observations of the reviewer are well appreciated. The specific responses are given below.

Comment

Use authors names, not citation numbers when citing them in the article. For example, line 325 states "49 reported" Should say "Fabreau et al. (49) reported…."

Response

Thanks for the important observation. This was an unintended error and had been rectifies in page 10 line 21 and 25.

Comment

Introduction should provide theme and purpose of the article, not jump right into the literature review.

Response

Thanks for this very important input.  This is now provided in Section 1 page 2 lines 44-51.

Comment

More information needed to define a descriptive study design.

Response

The comment is well appreciated. More information on the study design/instrument is provided in Section 2.3 page 7.

Thank you very much.

Reviewer 4 Report (New Reviewer)

The study aims to compare the socio-demographic characteristics, pattern of COVID-19 and healthcare attendance and utilization among migrant workers and non-migrants in the Kan-chanaburi Province in Thailand. Although I think the paper worth to be published, Authors need to improve the presentation of the results:

- Abstract: it doesn't need to be so detailed, so work on it to make it coincise and more focused on the real research question of the paper and the most important results;

- Materials and Methods: map is irrilevant as much in Table 1 the disaggreation of results by subprovinces if you don't comment further this disaggregation. Same for age disaggregation in Table 2: if you present that, which is the added value in your analysis?

- Discussion and Conclusions: you quote in the Abstract the importance of health insurance schemes for migrants in Thailand given the exclusion of health care provision for that portion of citizens but then neither in the Discussion nor in the Conclusion the public health topic becomes relevant. If you need to better link that two parts togheter to help Readers understand your point.

Author Response

Dear Reviewer 4,

The comments and observations of the reviewer are well appreciated. The specific responses are given below.

Comment

Abstract: it doesn't need to be so detailed, so work on it to make it concise and more focused on the real research question of the paper and the most important results.

Response

Thanks for this helpful comment. The abstract has accordingly been reviewed.

Comment

Materials and Methods: map is irrelevant as much in Table 1 the disaggregation of results by sub-provinces if you don't comment further this disaggregation. Same for age disaggregation in Table 2: if you present that, which is the added value in your analysis?

Response

The above comments are well appreciated. However, the relevance of the map (figure 1) is to show the geographical location of the study area.

Some more observations on table 1 have been provided in page 7 line 37-39.

A more statistical analysis of table 2 (new) has been performed in page 8 lines 24-27 and Table 2.

Comment

Discussion and Conclusions: you quote in the Abstract the importance of health insurance schemes for migrants in Thailand given the exclusion of health care provision for that portion of citizens but then neither in the Discussion nor in the Conclusion the public health topic becomes relevant. If you need to better link that two parts together to help Readers understand your point.

Response

The comments are well appreciated. A more statistical analysis of table 2 (new) has been performed in page 8 lines 24-27 and Table 2.

The need for health insurance for migrant’s health is mentioned in Section 4.2 page 11 in lines 40-43.

Thank you very much.

Reviewer 5 Report (New Reviewer)

This article looks at the differing usage of healthcare for migrants and non-migrants suffering from COVID-19.  The research question is an important question as healthcare for immigrants is a key social issue in destination regions.  With COVID-19 being one of the most significant and tracked healthcare issue in recent memory, this is a good source of data to use to demonstrate how healthcare opportunities are allocated to migrants and how they access these services.  

My only suggestion is to include a measure of significance for the demographic variables in table 2.  I think it would be useful to the reader to see if COVID infection rates differ between similar demographic groups of migrants and non-migrants.  This might also help explain the differing usage of healthcare between migrants and non-migrants.  

Overall, I think the paper is well researched and well written.  The topic is very important and the authors do a fine job of situating their research into the larger academic discourse.  I enjoyed reading the paper.  

Thank you for the opportunity to review your paper.  Best of luck in future research.  

Author Response

Dear Reviewer 5,

The comments and observations of the reviewer are well appreciated. The specific responses are given below.

Comment

My only suggestion is to include a measure of significance for the demographic variables in table 2. I think it would be useful to the reader to see if COVID infection rates differ between similar demographic groups of migrants and non-migrants. This might also help explain the differing usage of healthcare between migrants and non-migrants.

Response

The comments are well appreciated. A more statistical analysis of table 2 (new) has been analysed using Chi-Square in page 8 lines 24-27 and Table 2.

Comment

Overall, I think the paper is well researched and well written. The topic is very important and the authors do a fine job of situating their research into the larger academic discourse. I enjoyed reading the paper.

Response

Thanks for the compliment and helping to further enrich the paper for your useful comments and suggestions.

Thank you very much.

This manuscript is a resubmission of an earlier submission. The following is a list of the peer review reports and author responses from that submission.

Round 1

Reviewer 1 Report

the quality of the manuscript has been considerably improved.

Thanks for sharing.

Author Response

Dear Reviewer

COMMENTS FROM THE REVIEWERS AND THEIR RESPECTIVE RESPONSE/ACTION

Comment

The quality of the manuscript has been considerably improved.

Response

Thank you for the complement. This was achieved with your constructive comments and suggestion. We are indeed grateful for your contribution.

Sincerely,

Authors

Reviewer 2 Report

Dear Authors, 

Many thanks for taking time out to respond to my earlier comment. Please consider the following: 

i. I find the title a bit confusing, without reading the content, the title reads as though it is a comparative study between migrants and healthcare 

ii. the study contain several statistical data reported, however there was no mention of how this was analysed with the exception of the statement made in line 216-217.

iii. The result need to be more explicit and focused on the study theme 

iv. There is also need for proofread of the work

Author Response

Dear Reviewer

COMMENTS FROM THE REVIEWERS AND THEIR RESPECTIVE RESPONSE/ACTION

Comment

Many thanks for taking time out to respond to my earlier comment. Please consider the following: I find the title a bit confusing, without reading the content, the title reads as though it is a comparative study between migrants and healthcare.

Response

Thank you for this analytical view. However, migrants are the study population and ‘HEALTHCARE’ in this context is used to refer to ‘The treatment, diagnosis, amelioration, or prevention of sickness or illness ‘of migrants in that setting during the study period.

We are also open to a rephrase that can improve on clarity.

Comment

The study contain several statistical data reported, however there was no mention of how this was analysed with the exception of the statement made in line 216-217.

Response

We are grateful for this observation. This was an omission on our part. The analysis of the statistics was done using SPSS version 26 and provided in a descriptive summary statistic in tables and a graph. This has been clarified in page 6 lines 216-218

Comment

The result need to be more explicit and focused on the study theme.

Response

We are appreciative of this important observation as it helps with clarity and comprehension. The study generated epidemiological data and healthcare utilization of migrants in relation to COVID-19. To the best of our knowledge summary statistics provided in tables 1, 2 and 3 and the graph (figure 2) are suitable. Also, the discussion section has been divided into two sections to mainly focus on these issues as suggested by another respected reviewer. DISCUSSION (4.1. Migrants and vulnerabilities to COVID-19. 4.2. Healthcare access and utilization). This is to focus on the two main issues discussed in the article, COVID-19 vulnerabilities, and utilization and of healthcare by migrants. Page 9 line 299 and page 10 line 350.

Comment

There is also need for proofread of the work.

Response

Thank you for this all-important suggestion that helps to enhance the English Language. Proofreading has been done to correct errors and improve on the language quality.

Sincerely,

Authors

Reviewer 3 Report

In the discussion, you may divide them into several paragraphs and each paragraph should contain sub-headings, which can enhance readability.

Author Response

Dear Reviewer

COMMENTS FROM THE REVIEWERS AND THEIR RESPECTIVE RESPONSE/ACTION

Comment

In the discussion section, you may divide them into several paragraphs and each paragraph should contain sub-heading sections which can enhance readability.

Response

Thanks for this brilliant suggestion. Accordingly, the discussion section is divided into 2 parts under sub-headings: (4.1. Migrants and vulnerabilities to COVID-19. and 4.2. Healthcare access and utilization). This is to focus on the two main issues discussed in the article, COVID-19 vulnerabilities, and utilization of healthcare by migrants. Page 9 line 299 and page 10 line 350.

Sincerely,

Authors